# Glacial Water: A Dynamic Microbial Medium

**DOI:** 10.3390/microorganisms11051153

**Published:** 2023-04-28

**Authors:** Gilda Varliero, Pedro H. Lebre, Beat Frey, Andrew G. Fountain, Alexandre M. Anesio, Don A. Cowan

**Affiliations:** 1Centre for Microbial Ecology and Genomics, Department of Biochemistry, Genetics and Microbiology, University of Pretoria, Pretoria 0002, South Africa; pedro.lebre@up.ac.za; 2Rhizosphere Processes Group, Swiss Federal Research Institute WSL, 8903 Birmensdorf, Switzerland; beat.frey@wsl.ch; 3Departments of Geology and Geography, Portland State University, Portland, OR 97212, USA; 4Department of Environmental Science, iClimate, Aarhus University, DK-4000 Roskilde, Denmark; ama@envs.au.dk

**Keywords:** glacial microorganisms, glacier, meltwater, proglacial environment, water residence time

## Abstract

Microbial communities and nutrient dynamics in glaciers and ice sheets continuously change as the hydrological conditions within and on the ice change. Glaciers and ice sheets can be considered bioreactors as microbiomes transform nutrients that enter these icy systems and alter the meltwater chemistry. Global warming is increasing meltwater discharge, affecting nutrient and cell export, and altering proglacial systems. In this review, we integrate the current understanding of glacial hydrology, microbial activity, and nutrient and carbon dynamics to highlight their interdependence and variability on daily and seasonal time scales, as well as their impact on proglacial environments.

## 1. Introduction

Although the movement of glaciers and ice sheets profoundly affects the landscape, the most immediate protagonist of glacial influences is glacial meltwater [1]. Meltwater accelerates the movement of glaciers, evacuates glacially eroded sediment downstream, contributes to alpine hydrology and riparian zones, regional runoff, and ultimately sea level rise [2,3]. From a microbial perspective, meltwater transfers nutrients and microbes across and within the glacial mass [4,5] and eventually to proglacial ecosystems [6,7,8]. Flow systems within glaciers are complex and poorly known, varying spatially and temporally and differing between glaciers and ice sheets [9,10]. This spatial and temporal variability leads to continuously changing biochemical activities, nutrient cycling, and microbial community dispersion [4,11,12]. Microbial communities adapted to these icy environments transform meltwater chemistry, including both organic and inorganic compounds [11,13,14], and enrich glacial exiting waters with bioavailable molecules [13,15,16], making glaciers and ice sheets low-temperature bioreactors. Glacial microorganisms transform the geochemistry of glacial meltwater through a variety of metabolisms. For example, the microbial communities on the glacial surface are dominated by photoautotrophs and nitrogen fixers, whereas microbial communities in the subglacial waters can be dominated by iron reducers, sulfide oxidizers, sulfate reducers and methanogens [17,18]. Which metabolisms are most active at a specific time and in a specific glacial compartment highly depends on the state of the ice body [19,20]. Such data clearly indicate that glaciers and ice sheets are not inert masses, placing them as integral components of global biogeochemical cycling.

Ice sheets and glaciers occupy 10% of the Earth’s surface and represent about 68% of Earth’s freshwater [21]. Their retreat, and the consequent increase in released meltwater, will have global impacts [22,23,24,25]. Meltwater discharge from glaciers and ice sheets is enormous; the Greenland Ice Sheet alone is expected to release 357 ± 58 Gt water y^−1^ [26]. Meltwater discharges to a wide range of different proglacial systems, from glacial forefields to glacial lakes or oceans, impacting these environments by contributing microorganisms, nutrients, and sediment [6,27,28]. Cameron et al. (2017) estimated the export of 6.9 × 10^22^ cells y^−1^ from Greenland alone [6], and about 1.1 × 10^22^ cells y^−1^ are expected from its surface under a medium emission scenario in the next 80 years [29]. This export of nutrients and microorganisms to proglacial systems is, therefore, thought to represent a major contribution to global biogeochemical cycling in proglacial areas [27,30,31,32,33].

Although several recent reviews of the microbiomics of glacial systems are available [4,34,35], this review provides an up-to-date holistic view of the interplay among glacial hydrology, microbial activity, and geochemistry. In fact, through an extensive literature review, we compile the current state of knowledge on glacial hydrology, microbial activity, and geochemical cycling; we explore their interdependence and how differences in glacial meltwater flow systems influence nutrient distributions and microbial processes. In this review, we also explore how glacial processes and patterns influence a variety of proglacial environments, outlining the possible global impacts of increases in meltwater discharge due to climate warming.

## 2. Glacial Hydrology

Glacial hydrological systems consist of interconnected micro- and macro-pathways, their size and location being influenced by the glacial thermal regime (Box 1) and water availability [36], which, in turn, are related to climate (e.g., latitude, altitude, wind, precipitation, temperature, and radiation) [37,38].

Box 1Ice temperature and permeability. Warm atmospheric temperatures and high precipitation rates lead to temperate ice (i.e., ice temperature above the ice melting temperature), where snow deposition can isolate surface ice from constant subzero temperatures. On the contrary, surface cold ice (i.e., ice temperature below the ice melting temperature) is the result of permanent cold temperatures [39]. Whereas surface glacial ice is directly influenced by the atmospheric temperature, the temperature of the subglacial ice is influenced only indirectly: i.e., the temperature of subglacial ice is conditioned by geothermal heat fluxes and by the amount of heat that is generated by glacial sliding, glacial water flow, and ice deformation, which are deeply conditioned by the supraglacial water dynamics [40]. Ice temperature profiles are used to classify glaciers. Temperate glaciers are those entirely composed of ice with a temperature above the ice melting temperature (i.e., temperate ice), cold glaciers are composed of ice with a temperature below the ice melting temperature (i.e., cold ice), and polythermal glaciers have a more complex thermal structure characterized by different ice temperatures [39]. Polythermal glaciers often consist of temperate ice throughout the glacier body, except the first meters of the glacial surface (typically in the ablation zone), where the ice is directly in contact with the cold atmosphere and maintains a cold temperature across its surface ice. Examples of temperate glaciers can be found between alpine glaciers (e.g., Arolla glacier), polythermal glaciers between Arctic glaciers (e.g., Midtre Lovénbreen), and cold glaciers between polar glaciers (e.g., Larsbreen) [39,41]. Ice temperature influences glacier and ice sheet functioning, where small changes in ice temperature can deeply change the glacier dynamics [42,43]. For instance, ice temperature determines how permeable the ice is [1]. Primary permeability is the movement of water at a small scale through ice veins and only occurs in temperate ice. The volumetric water content in temperate ice has been estimated to reach 9% of the ice volume [44]. On the contrary, cold ice is not permeable and therefore does not present a liquid matrix between ice crystals [1]. Contrary to temperate glaciers, water only flows through crevasses, moulins, englacial channels, and conduits (secondary permeability) in cold glaciers [45,46]. Whereas an active englacial system (i.e., with flowing water) has been observed in a wide variety of glacial thermal regimes [36,47,48], temperate glaciers have the most developed water discharging system compared to the other glacier types due to the higher permeability and malleability of temperate ice [1]. Differences in permeability, therefore, lead to the development of different water discharge networks: e.g., because of the less developed englacial channel network in cold glaciers, water is discharged mainly through the supraglacial environment in most systems during the summer [49], where supraglacial features are more developed compared to those in the other ice systems [50].

### 2.1. Seasonal Variation

The presence of water in glaciers is highly seasonal, as the generation of glacial water depends on the energy available to melt snow and ice. Water input also comes from rain events. For most glaciers in the temperate and subpolar regions, mass accumulates during the accumulation season (e.g., winter), which is characterized by colder temperatures and snow accumulation [1]. By comparison, mass loss, including melt, occurs during the ablation season (i.e., summer), which is characterized by warmer air temperatures and greater solar radiation. During this season, the lower elevation ablation zone of a glacier is gradually exposed as the winter snow cover melts, revealing an ice surface. In the accumulation zone, located at a higher altitude on the glacier, only some of the accumulated snow melts, therefore accumulating and compacting it into firn (transition stage between snow and ice) and ultimately into ice. The amount of snow gained over winter minus the amount of snow and ice lost defines the glacial ice balance [1].

#### 2.1.1. Ablation Season

In spring, the snowpack temperature increases through heat conduction from warm air and via refreezing of meltwater that percolates into the cold snow (Figure 1). Once the snowpack becomes isothermal at 0 °C, snow melt percolates into the entire seasonal snowpack. In the accumulation zone, the meltwater enters the firn, forming a firn aquifer that drains to nearby crevasses [51,52,53] (Figure 2). In the ablation zone, the water accumulates on the ice surface and also drains into nearby crevasses [54]. Once the snow melts off the ice in the ablation zone, a shallow weathering crust of porous ice forms, caused by penetration of solar radiation and enhanced by the flow of meltwater and potentially warm air [55]. The resulting weathering crust represents a photic zone a few centimeters to a few meters deep [55,56,57,58]. Both the weathering crust and the firn aquifers are active parts of the glacial hydrological system and contribute to accumulating, distributing, and regulating water discharge to the englacial system [36,51,52,53,55,57,59,60]. Cryoconite holes can also form on the ice surface as a result of the deposition of sediment and biological materials on the glacial surface. These patches of material melt into the ice faster than the surrounding ice, driven by increased heat absorption by the dark material [61,62,63]. Under the right circumstances, these holes can form glacial ponds and lakes (cryo-lakes) [39,50]. Thus the surface streams, weathering crust, cryoconite holes, and cryo-lakes form the complex surface hydrology of glaciers [64]. This supraglacial hydrology influences the drainage pattern and flux of meltwater into the glacial interior (i.e., englacial system) and bed and to proglacial systems [64].

The main pathways from the glacier surface and firn aquifers to the glacial interior are moulins and crevasses. Moulins constitute a direct connection to an englacial system of conduits, whereas crevasses drain water via a network of fractures thought to connect to englacial conduits [69,70,71]. The englacial system of conduits reaches the bottom of the glacier, where it flows along the ice-substrate interface and, if the substrate is sediment, through the sediment as well.

Most subglacial water originates from the glacier surface, although a small flux may originate englacially and subglacially due to frictional dissipation of heat generated by flowing water, ice deformation, or from geothermal heat at the bottom of the glacier [72,73,74]. The Antarctic ice sheet is a major exception, where the main source of subglacial water is basal melt [75]. Generally, subglacial water flows in two main types of systems: slow flow within a distributed network of linked cavities or confined groundwater flow within a layer of subglacial rock debris (i.e., subglacial till), or in a quick flow system composed of a network of subglacial channels [76,77]. Distributed systems are thought to be water-filled all year round, whereas channels are water-filled only at high meltwater discharge [11,36,78]. Under ice sheets, subglacial lakes may occur as stable subglacial water bodies that can be isolated or hydrologically connected to a subglacial flow system [79,80]. Regardless of its path and residence time within the glacier, water will eventually escape to proglacial systems via streams or groundwater flow.

#### 2.1.2. Seasonal Evolution

During summers with plenty of meltwater, surface streams flow over the ablation zone, and the weathering crust, including cryoconite holes, is full of water [1] (Figure 1). Surface water that accumulates in weathering crust and cryoconite holes drains into crevasses and moulins, and water in surface streams commonly pours into moulins. Compared to supraglacial systems, water accumulated in firn aquifers is less connected to the rest of the glacial system, and it occasionally drains into nearby crevasses [54]. Englacial and subglacial channels (quick flow) are thought to be filled when water discharge reaches its daily maximum and are less full early the following morning when the air temperature is the coolest, and the sun is just beginning to warm the snow and ice [81,82]. The subglacial distributed systems are thought to be continually water filled [11]. In addition to daily cycles of glacial runoff, longer variations occur with the occurrence of warm and cool weather systems [81,82]. Episodic floods may also occur due to the sudden release of ponded water within the glacier or to the sudden drainage of glacier-dammed lakes [83,84,85].

At summer’s end and the beginning of winter’s accumulation season, much less water is present in the glacial system due to the low air temperatures and reduced solar radiation, which largely eliminates surface melt, and while precipitation falls mostly as snow [1]. Some water may remain unfrozen over winter in subsurface weathering crust and firn aquifer due to the insulating effects of a deep snow cover [63]. Over winter, the englacial and subglacial channels collapse due to ice pressure and the absence of a counteracting pressure from flowing water [36,86,87]. However, due to a decrease in water flow, water pockets may form within the englacial or subglacial system [88,89]. Glacial streamflow also often continues through the winter, albeit at very low discharge. This may be due to the drainage of water resident in the glacier as well as meltwater produced by geothermal heat and by deformation within the ice [54,81].

### 2.2. Hydrological Residence Times

The residence time of water in various parts of the glacial hydrological system varies (Table 1) and influences in situ geochemical processes [90]. The extent to which microbial communities can develop and contribute to geochemical modification depends both on microbial doubling times [31,58] and on how long they can reside in a specific glacial compartment (i.e., water residence time). Furthermore, different residence times create a variety of conditions for microbial metabolism [5,12]. For example, the lower the residence time, the better ventilated a system is with the consequent creation of oxic waters. Oxic waters will promote aerobic or facultative anaerobic microbial metabolism, such as nitrification and iron oxidation, whereas anoxic waters will promote the growth of anaerobic organisms, such as methanogens and sulfate-reducing bacteria and archaea [19].

The residence times of water in a glacier depend on the glacier’s size, where the bigger the glacier, the longer the water takes to flow through the system because of longer routing pathways [67]. Meltwater residence in ice sheets is, therefore, generally higher than in glaciers [9,81]. In the photic zone on and just below the glacier surface, water residence times in the weathering crust are at least several days during the ablation season [58,60]; weathering crust can store some water in winter if covered with a thick snowpack [55]. Ice-lidded cryoconite holes found on ice sheets and polar glaciers store water from days to months, and isolated cryoconite holes that melt within the ice and refreeze annually without connecting to the surface or subsurface hydrologic system may store water for more than a decade [61,91,103]. Open cryoconite holes on temperate and polythermal glaciers are connected to the supraglacial hydrological system and may have residence times of a few minutes to hours [39,63]. Water residence times in supraglacial lakes and ponds vary similarly to those of cryoconite holes [50,61,64]. Supraglacial streams are usually fast water-flowing systems where water fluxes vary based on the glacial system state [64].

Below the photic zone, within the firn aquifer of the ablation zone, water residence times can range from hours to days [54,104]; perennial firn aquifers have also been observed [51,52,92,93,94,105]. Within the englacial realm, two very different water-flowing systems exist. A quick flow system of conduits may have residence times of hours and perhaps up to a day [88,96,97]. A very slow system exists along the boundaries of the ice crystals, where three or more crystals meet, creating a small flow passage (i.e., ice vein) [98]. In ice veins, water flows at very slow rates and is easily blocked by air bubbles. However, these veins can host viable microorganisms [106]. Residence time in these veins is unknown. At the bottom of the glacier, channels residence time can be hours, whereas, in slow-flow distributed subglacial systems, it can be days to months [99,100,101], reaching estimated water permanence times of millions of years in subglacial lakes under ice sheets [79,80,102].

As with meltwater discharge, water residence times change seasonally, with the shortest residence time being in the summer [90]. In winter, after the system closes down and various components are no longer linked due to the lack of water flow, the freezing of passageways or ductile closure of passageways causes residence times to increase [90].

## 3. Hydrology Influences on Glacial Nutrients and Microbial Communities

### 3.1. Deposition of Nutrients and Microorganisms on Glacial Surfaces

Dry and wet deposition of atmospheric aerosols transports diverse chemistry and biology to the glacier surface [107,108,109,110,111]. Their chemistry and concentration depend on patterns of atmospheric circulation, distance to source regions, and type of source emissions [112,113,114]. The distribution of chemical compounds across a glacier can be heterogeneous due to variable aeolian deposition [115]. For example, mineral dust is more highly concentrated at the margins of a glacier due to its proximity to bare rock and soils [116].

Bioaerosols (particulates containing microorganisms) may inoculate glacier surfaces, and subsequent microbial community development depends on the surface environment and microbial adaptations [117,118,119]. Such microbial communities typically include both endemic and cosmopolitan microorganisms [120,121,122,123,124,125]. Considering the dynamic and ever-shifting nature of glacial ice sheet systems, these communities will experience a broad range of icy micro-environments and will shift accordingly as they transit through the system [34,35].

### 3.2. Microbial and Geochemical Dynamics during the Ablation Season

#### 3.2.1. Supraglacial Realm

In the supraglacial environment, organic and inorganic nutrients and carbon are often available in dissolved forms released by the biochemical weathering of deposited particles [13,126]. With the onset of the ablation season, nutrients and carbon deposited during the accumulation season progressively percolate through the snowpack into the weathering crust, cryoconite holes, and glacial ponds (ablation zone) [12,55], and microorganisms are able to resume metabolic activity shortly after thawing [20] (Figure 1).

In the photic layer of the weathering crust, UV- and visible-light-driven chemical transformations lead to the dissolution of iron oxide and silicates in mineral particles and Fe^3+^ reduction in ice-hosted sediment particles [13,55,127]. Nutrient concentrations, particularly dissolved organic nitrogen (DON) and P are generally higher in weathering crust compared to supraglacial streams and cryoconite holes [128]. Cryoconite holes are widely recognized as hotspots of microbial activity [63,91,129,130] (Figure 2). The waters are generally oxic environments, and ice-lidded cryoconite holes in Antarctica can be supersaturated in O_2_ [103,131]. However, within thick cryoconite granules (aggregates of microorganisms and organic and inorganic nutrients present in both open and ice-lidded cryoconite holes), the environment can become anoxic [132]. The oxidation state of cryoconite holes is important as it dictates the oxidation state of key nutrients. For example, the most common Fe ions, Fe^2+^ and Fe^3+^, are soluble only within certain pH and dO_2_ conditions, affecting their capacity to associate with other ions (e.g., chloride and hydroxide ions) or to adsorb onto ice crystal surfaces [13].

With the onset of the ablation season, during which the primary nutrient input is from snowmelt, the exposed component of the supraglacial environment (e.g., weathering crust) is dominated by prokaryotic photoautotrophs (e.g., Cyanobacteria in cryoconite holes) and ice algae (e.g., *Ancylonema nordenskiöldii*, and *Mesotaenium berggrenii* on the surface of the ice/weathering crust) [111,133]. These organisms can directly affect melt rates and surface morphology (Box 2) and play an important role in fixing atmospheric CO_2_ and N_2_ when nitrogen-fixing cyanobacteria, such as *Nostoc* and *Anabaena*, are present [134,135]. Microbial activity changes as the ablation season progresses (Figure 1). For example, during initial phases of snowmelt early in the ablation, excess concentrations of inorganic nitrogen (NO_3_^−^ and NH_4_^+^) flush from the snowpack and are utilized by microbial communities, while later in the season (after the inorganic flush and when concentrations are low) microbial activity switches to dinitrogen fixation [136,137,138]. Phototrophic communities help to produce dissolved organic nutrients (i.e., dissolved organic carbon, nitrogen, and phosphorous), enriching the glacial surface with organic and inorganic nutrients that would be otherwise limiting factors for heterotrophic microbial activity [124,128,139]. Phosphorous is typically sourced from supraglacial particles via geochemical and physical processes and from microbial necromass via biological activity [139,140]. Microbial exudates and necromass on the glacial surface are essential for the functioning of the heterotrophic component of the microbial community. In this context, it has also been observed that different exudates are differentially utilized by microorganisms [141,142] within a complex glacial microbiome where heterotrophic organisms present a wide range of metabolisms [17,49,142]. Bradley et al. (2023) observed that more than 50% of bacterial cells (dominated by Actinomycetota, Pseudomonadota, and Planctomycetota) are translationally active on glacial surfaces [20]. Most of the deposited and transformed nutrients in the supraglacial realm are then exported to the rest of the glacial system [31,35]. With the development of warmer conditions due to the progression of the ablation season, weathering crust and cryoconite holes contribute nutrients and microbial cells to the rest of the glacier [29,58,143,144].

Box 2Albedo and bioalbedo: how nutrients and microorganisms influence glacial hydrology. Although the subject of this review is how glacial hydrology influences nutrient and microbial distribution, the contrary is also true. The distribution of nutrients and microbial cells influences supraglacial water flux. Albedo (i.e., the proportion of light that is reflected by a surface) is lower for darker surfaces compared to lighter surfaces. Therefore, higher input of mineral dust and black carbon on the glacial and ice sheet surfaces decreases the albedo, causing higher heat absorption and increased melting, which influences the morphology of the ice surface [145]. Bioalbedo is a new term created to specifically refer to the decrease in albedo provoked by snow and ice algae, which are dark-pigmented, on the glacial surface [146,147]. A change in albedo and bioalbedo, therefore, directly influences glacial hydrology by promoting glacial and ice sheet surface melting.

#### 3.2.2. Englacial Realm

Snowmelt also enters into the firn aquifer. No studies reporting biological processes in firn aquifers are available. However, there are indications of microbial activity: Holland et al. (2022) observed similar NO_3_^−^ concentrations between the ice-snow interface meltwater and the snowpack [128]. However, DON and NH_4_^+^ concentrations were variable, possibly indicating microbial activity at the ice-snow interface. The chemical composition of the meltwater that enters the englacial system varies across the ablation season. Whereas meltwater at the beginning of the ablation season is likely to reflect the chemical composition of the snow, microbial processes and particle weathering ensure that meltwater that enters the englacial system later in the season is enriched with carbon, macronutrients (e.g., N, P, and Si) and ions such the dissolved inorganic forms of sulfur (e.g., SO_4_^2+^) [148,149]. The englacial realm, with its network of water pathways, transfers cells and nutrients within the glacial system. It is unclear whether the englacial realm also has a role in nutrient and carbon transformation and is characterized by a microbial community specific to englacial pathways conditions [150]. Microbial nutrient cycling observed in fast-flowing supraglacial streams [49] suggests that microbial processes may be significant in fast-flowing englacial conduits. In addition, ice cores collected from englacial systems indicate that microorganisms are not quiescent but maintain an active metabolism [151]. These active metabolisms could be ascribed to chemoautotrophic organisms [151] or could rely on simple carbon substrates (e.g., acetic and formic acids) using NO_3_^−^ and SO_4_^2+^, which are abundant in meltwater deriving from supraglacial systems, as terminal acceptors [19].

In addition to englacial conduits, ice veins, with their high nutrient concentrations, offer a favorable habitat for microbial activity [68,106,152,153], and high cell concentrations have been measured in this environment [154]. Liquid flow in veins has long residence times, suggesting low oxygen concentrations and anaerobic metabolism [68,155], which can be mediated by methanogens such as *Methanosphaerula* and *Methanococcus* [156]. Despite the high nutrient concentrations and favorable conditions for active communities, microbial structure and function in the englacial realm are poorly understood [150,151].

#### 3.2.3. Subglacial Realm

In subglacial systems, meltwater contacts bedrock (and subglacial till) and is consequently enriched in compounds released by rock comminution and dissolution (e.g., H_2_ and FeS_2_), creating environments where redox conditions may vary widely [65,66]. Chemical compositions of subglacial water highly vary due to (i) supraglacial and englacial hydrology, which controls the pattern, discharge, and biogeochemistry of meltwaters reaching the bottom of the glacier; (ii) subglacial hydrology, which controls the discharge and residence time of waters along the bed and influences patterns of erosion and regelation; and (iii) the geology of the subglacial substrate [11,90,121,143,157].

Depending on the mineral composition of the glacier and ice sheet beds, subsurface meltwater can be influenced by the weathering of pyrite (i.e., pyrite oxidation) and/or carbonates (i.e., carbonate dissolution) [27,158]. The dissolution of pyrite releases protons, and the dissolution of carbonate rocks releases dissolved inorganic carbon (e.g., CO_2_), which then creates carbonic acid in aqueous environments [18]. Pyrite oxidation is the prevalent form of mineral weathering in subglacial environments and has been observed to drive subglacial microbial metabolism [18,157]. The acidic environment resulting from pyrite oxidation also drives carbonate and silicate weathering [159]. The presence of subglacial tills also influences meltwater chemical compositions [36,160,161], and glacial beds can also be connected via aquifers to subterranean water sources [162]. All these factors further influence meltwater chemical composition and shape microbial community input to the subglacial environment [11,162,163].

As in the supraglacial and englacial realms, subglacial microbial communities are largely composed of heterotrophic microorganisms. However, contrary to the supraglacial realm, the primary producers of subglacial communities are chemolithotrophs. These organisms rely on nutrients transported from the glacial surface but also those released by rock weathering, and they accelerate mineral weathering of the glacial bed and chemical transformations within the subglacial system [13]. Biotic pyrite (FeS_2_) dissolution is rapid both in oxic and anoxic conditions [13,164], where O_2_ and Fe^3+^ can be used as sulfide oxidants [66]. Sulphide oxidation in oxic conditions uses pyrite, oxygen, and water to produce H^+^, Fe(OH)_3_ (iron (oxyhydr)oxides), and SO_4_^2^. Fe(OH)_3_ dissociates to Fe^3+^ in the acidic subglacial environment (created by a high concentration of H^+^ due to rock dissolution) [13]. Anoxic pyrite dissociation can then occur: pyrite, Fe^3+^, and water react to form Fe^2+^, SO_4_^2−^, and H^+^. Anoxic pyrite dissociation is faster than oxic dissociation because of the higher H^+^ production, which accelerates rock dissolution and weathering [13,165]. These weathering reactions are mediated by iron/sulfur-oxidizing bacteria, such as *Thiobacillus* and *Sideroxydans* species, and iron-reducing bacteria, such as *Desulfosporosinus*, *Geobacter* and *Rhodoferax* species [18,157,165,166]. Other microbial-mediated processes in the subsurface environment include denitrification, Mn^4+^ reduction, SO_4_^2−^ reduction, methanogenesis, and nitrification [19,108,167,168,169]. In this environment, complex microbial interactions occur where, under anaerobic conditions, SO_4_^2−^ reducing bacteria compete with methanogens for carbon substrates [170].

While meltwater that reaches subglacial systems is mostly oxygenated, as it is sourced from well-ventilated environments (e.g., supraglacial streams, moulins, and fast-flowing englacial conduits), oxygen levels in the subglacial environment can significantly vary based on the morphological characteristics of the system and, consequently, on water residence times [11,159]; whereas channelized subglacial systems are characterized by oxic waters thanks to their fast-flowing waters [11], anoxia and higher rock dissolution rates are observed with longer water permanence times and typically indicate higher rates of microbial activity and respiration [11,159]. In distributed drainage systems, there is a progressive development of anoxic conditions due to slow water flow, which creates favorable conditions for the uptake and use of organic matter by heterotrophic organisms via oxidative cellular respiration [67]. Despite these shifts in water oxygen levels, in general, microbial metabolism is thought to be driven by the mineralization of organic carbon and nitrogen under oxic conditions [19], whereas microbial communities performing sulfate reduction and methanogenesis prevail with the development of anoxic conditions [171].

### 3.3. Microbial and Geochemical Dynamics during the Accumulation Season

Most of the studies on glacial microbial communities take place during the summer ablation season at temperate and polythermal glaciers, where geochemical and microbial processes are most active due to the presence of meltwater and nutrients in the system [172]. Consequently, little information is available on microbial processes during winter.

Even in winter, water in ice veins, englacial pockets, and subglacial regions [11,19,68] may retain sufficiently high solute concentrations to sustain basal microbial metabolism (e.g., DNA repair mechanisms) [173] or even microbial growth. During the accumulation season, the prevalent microbial metabolism is likely to be chemoautotrophy in all glacial environments; the newly secreted nutrients and remnant nutrients from ablation seasons could also sustain metabolism in heterotrophic organisms. Active microbial communities have been identified in systems that are only minimally influenced by ablation/accumulation seasonal differences, such as subglacial lakes in ice sheets [174,175,176]. Furthermore, in glaciers during the accumulation season (when nutrient input from the surface is largely absent), nutrients can be sourced from glacial bedrock where H_2_ is released abiotically from rock comminution and can be oxidized in both aerobic and anaerobic conditions, fueling microbial chemolithotrophy [65,177,178,179]. Thus, even during the accumulation season, glaciers can harbor biogeochemical transformations thanks to microbial-mediated processes. This is also supported by observations of active heterotrophic communities in snowpacks incubated in cold and dark conditions [180].

## 4. Glaciers and Ice Sheets as Bioreactors

Considering the array of microbial processes active on, in, and under glaciers and ice sheets, these systems can be thought of as bioreactors. Nutrient concentration and bioavailability increase from supraglacial input waters to exiting subglacial waters. Exiting subglacial waters are enriched with dissolved iron (i.e., Fe^2+^ and Fe^3+^) and iron nanoparticulates (e.g., Fe (oxyhydr)oxides and Fe^2+^-bound compounds), which are bioavailable for microbial uptake [13,181]. Similarly, other metals are more bioavailable in glacial flour (i.e., fine rock particles formed by rock comminution) than in other dust [182]. Organic carbon (DOC) also follows similar trends [183]. The DOC proteinaceous component (DOC that is microbially produced and secreted) is higher compared to humic and fulvic acids (DOC transported by aerosol deposition and terrestrial transport) in glacial meltwater compared to other water systems such as rivers and lakes [16], indicating a strong involvement of glacial microbial communities in carbon cycling, and in the export of labile carbon. Kellerman et al. (2020) observed a shift in protein-like fluorescence with the progression of the ablation period indicating microbial communities as a source of DOC [184]. Barker et al. (2006) also showed that supraglacial and subglacial water had similar DOC concentrations but different fluorescence signals in three different glaciers, clearly indicating shifts in DOC quality due to microbial transformation [15]; this further points to a microbial role in DOM release [7,185]. An increase in microbial cell concentrations in glacial water flowing from the surface to the subsurface of glaciers has also been reported [6] (Table 2). Cell concentrations in subglacial water are usually in the order of 10^5^ cells mL^−1^ compared to concentrations in supraglacial water which are an order of magnitude lower (with the exception of ice veins where observed concentrations were 10^6^–10^8^ cells mL^−1^) (Table 2). These clear patterns in the enrichment of cell, nutrient, and bioavailable compounds show how glaciers serve as bioreactors for a range of biogeochemical transformations, which in turn can deeply influence proglacial systems.

## 5. Proglacial Systems

Glacial meltwater is an important source of cells and nutrients in outflow systems, carrying bioavailable Fe, DOC, N, P, Si, and rare metals, together with sediments and glacial flour [183,192,193,194,195,196,197]. How the proglacial system impacts downstream systems [198,199,200,201] depends on different factors, where the morphology of the proglacial system (i.e., land or maritime terminating) and its nutrient state (e.g., oligotrophic vs. eutrophic) play important roles [202].

The retreat of land-terminating glaciers exposes soil in the proximity of the glacier (i.e., forefield), where soils show a gradient in texture and chemical characteristics from the ice edge: newly exposed soils are usually characterized by low nutrient levels and little or no vegetation, and are impacted by microorganisms and nutrient from glacial meltwater [135,203,204,205]. Subglacial water from land-terminating glaciers also flows into rivers and streams, which will then export nutrients to other water bodies (e.g., lakes, rivers, and seas) [206]. Glacial streams and lakes typically show different dissolved inorganic nitrogen and phosphate concentrations than similar water bodies without glacial water input, and silica concentrations are lower in glacial streams compared to non-glacial streams [200]. Streams and lakes are also enriched with nitrogen deriving from glacial water in North America [207,208], where an increase in nitrogen concentration influences microbial structure and diversity (e.g., planktonic diatoms) [209]. Warner et al. (2017) also observed higher algal biomass in glacial lakes (c.f. non-glacial lakes) [199]. Lake connectivity to glacial water influences microbial communities due to the import of nutrients and the shift between turbidity and clear water conditions [210].

During the transport in the river, glacial meltwater undergoes chemical modifications. For instance, Fe and DOC are sequestered by precipitation and adsorption during the glacial meltwater transport in rivers and streams in Svalbard [211]. Nitrogen was also observed to decline in downstream glacial lakes in North America [199]. However, DOC increased during its flow in proglacial streams in Iceland, probably due to carbon mineralization and microbial transformation [212]. These contrasting results show how the processes in the proglacial streams are regulated by many local factors. Even if exported nutrient concentrations (e.g., Fe) decrease during their flow through the rivers and estuaries, glacial-fed streams are still able to export nutrients to the ocean [213]. Glacial meltwater from land-terminating glaciers enters the sea/ocean through estuaries. Here, the glacial water creates a top layer of sediment-rich water which can inhibit primary productivity in the proximity of the land because of the decrease of light filtration in the system [214,215] (Figure 3A). Whereas this is often the case in the Arctic, land-terminating glaciers in the Antarctic can have positive effects because water is so oligotrophic that even the import of low nutrient concentrations can increase primary production [202,213,216,217].

Subglacial water from marine-terminating glaciers provides carbon, macronutrients (e.g., nitrogen), and micronutrients (e.g., iron) to the surrounding marine system [214,218,219]. Organic carbon and other macronutrients are generally less concentrated in subglacial water than in seawater but more bioavailable [183,192,220]. In addition, glacial meltwater causes an upwelling of nutrient-rich deep seawater due to the buoyancy of the glacial cold water (Figure 3B), therefore enriching the surface seawater column with deep sea macronutrients [220,221,222,223]. Micronutrients (e.g., Fe) are generally more concentrated in subglacial waters compared to seawater [193], although the destiny and availability of these nutrients are still under debate [193,195,221].

The impact that subglacial discharge has on the coastal system depends on two main factors: the seawater depth at which subglacial water is discharged and the nutrient state of the marine system [224,225]. Meltwater from marine-terminating glaciers enters the marine environment at different depths in the seawater column depending on the kind of glacier (e.g., tidewater or ice shelf) and glacier thickness [225]. If the discharge of cold subglacial waters creates an upwelling of deep nutrient-rich waters that reach the photic zone of the seawater column, an increase in primary productivity (e.g., higher chlorophyll concentration) is observed thanks to the import of nitrogen, ammonium, phosphate, and silicate in the Arctic marine coastal environment where the seawater is generally nitrogen depleted [220,221,222,224]. However, the upwelling of nutrient-rich waters might not reach the photic zone in the case of deep or shallow marine-terminating glaciers, causing limited primary productivity in Arctic waters [225]. Contrary to the Arctic seawater, which is generally nitrogen depleted, the Southern Ocean is mainly limited by low concentrations of bioavailable Fe (e.g., Fe^2+^ and colloidal Fe) [224,225], and it, therefore, relies less on the inputs of macronutrients by deep seawater upwelling, but rather more on the import of micronutrients from glacial waters themselves [195,226,227].

Generally, nutrient export increases with meltwater export to the proglacial systems [148]. This is the result of the release of the nutrients trapped in glaciers and ice sheets, which act as storage for different elements such as carbon, and also of the higher presence of water in the system, which promotes geochemical and microbial processes in supraglacial and subglacial realms [183]. Whereas it is clear that an increase in nutrient release affects geochemical cycles, trophic chains, and microbial diversity [23,228,229], an increase in nutrient export from the glacial system does not always correspond to an increase in the proglacial system primary production [221,230].

Changes in water residence times in glaciers could cause drastic shifts in proglacial environments. In particular, a decrease in glacial residence times with a consequent increase of glacial water released to outlet systems is observed due to global warming [9,231,232]. For example, a change in the glacial water runoff and nutrient export can correspond to a change in water acidity in the coastal environment caused by a variation in carbonate concentrations, sediment input and burial, water stratigraphy and turbidity, and nutrient import into the system, leading to a change in environmental dynamics and primary productivity in proglacial systems [148,233,234,235,236,237,238] (Figure 3C).

## 6. Conclusions

Supraglacial and subglacial microbial communities have been widely studied and characterized in the last few decades, with many studies showing how biotic and abiotic processes in these systems are linked to global biogeochemical cycles [4,34,35]. It is also common knowledge that climate warming affects the interconnectivity among glacial hydrology, microbial community, and geochemistry and that increases in meltwater discharge from glaciers and ice sheets impact the proglacial systems [20,198,199,200,201]. Therefore, whereas it is clear that climate warming affects and will affect dynamics within glaciers and ice sheets and in their out-stream environments, little information is present on how glacial microbial communities will be impacted. In order to fully understand these impacts, we believe a more holistic understanding of how microbial communities, nutrient cycling, and glacial meltwater hydrology function and interact is necessary. For example, nutrient cycling and microbial activity in the yet unexplored englacial channels and firn aquifers should be explored, together with biotic and abiotic processes during the accumulation season. Further, better estimates of water residence times in different compartments of glaciers and ice sheets are needed to more confidently estimate microbial activity and conditions for nutrient cycling. Whereas more comprehensive information on glacial and ice sheet hydrology will inform on microbial activity and nutrient composition, the opposite is also true, where a deeper understanding of microbial and nutrient distribution and transformations could further inform on water paths within glaciers.

## Figures and Tables

**Figure 1 microorganisms-11-01153-f001:**
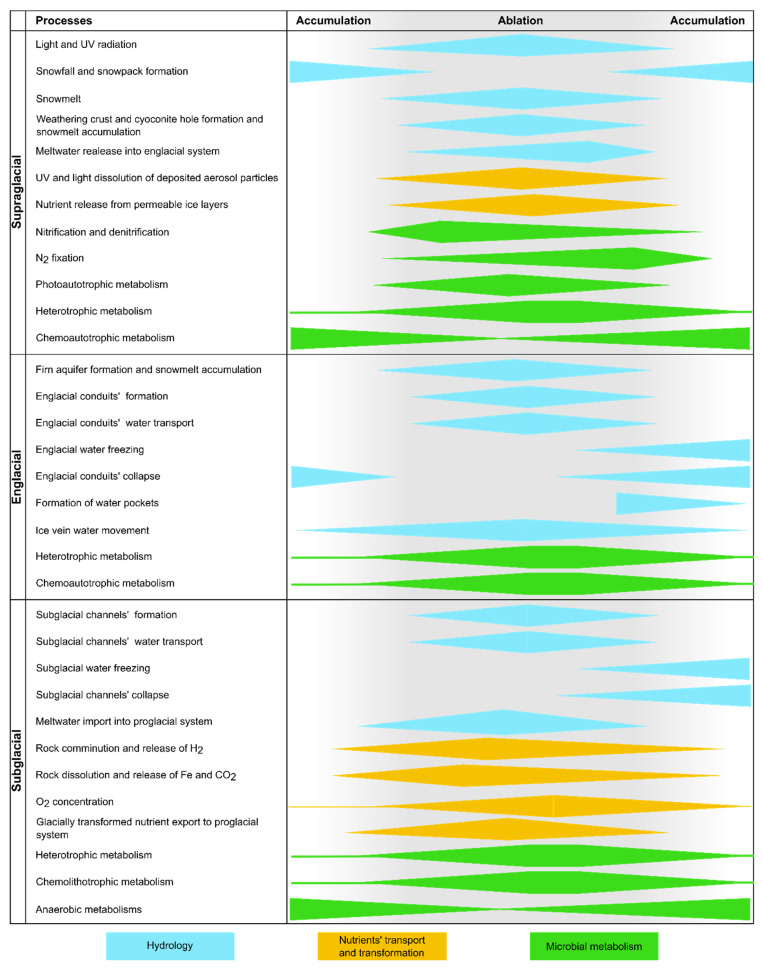
Scheme of hydrological, nutrient, and microbial metabolism shifts during ablation and accumulation season. Width and height of the colored shapes represent the occurrence and the intensity of each specific process. *X*-axis represents time (accumulation and ablation seasons), whereas the *y*-axis represents different glacial realms (supraglacial, englacial, and subglacial environments). Water release from weathering crust, cryoconite holes, and especially firn aquifers is not linear as it also goes through daily and irregular cycles. The proposed trends of hydrological [1,36,55,64], nutrient [13,65,66,67], and microbial metabolism [11,17,18,68] shifts are broadly based on published peer-reviewed studies.

**Figure 2 microorganisms-11-01153-f002:**
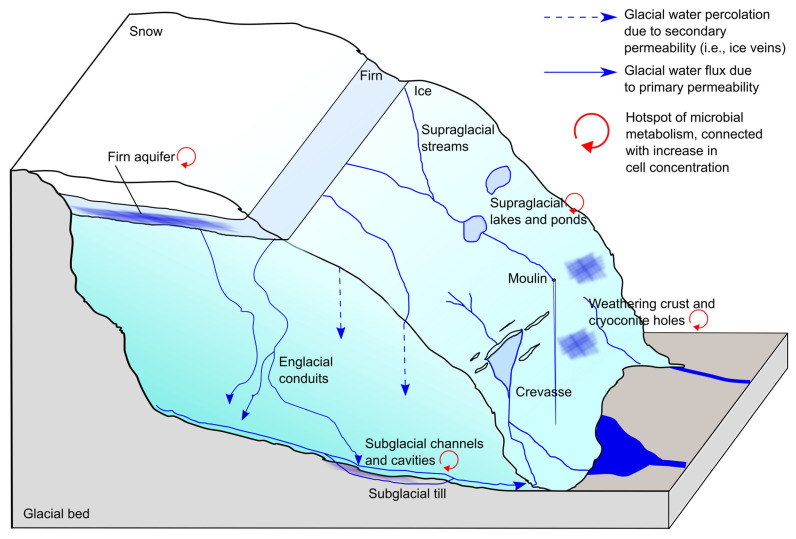
Water fluxes during the ablation season and main hotspots for microbial metabolisms. Subglacial lakes (not represented in this figure) can also represent hotspots for microbial activity.

**Figure 3 microorganisms-11-01153-f003:**
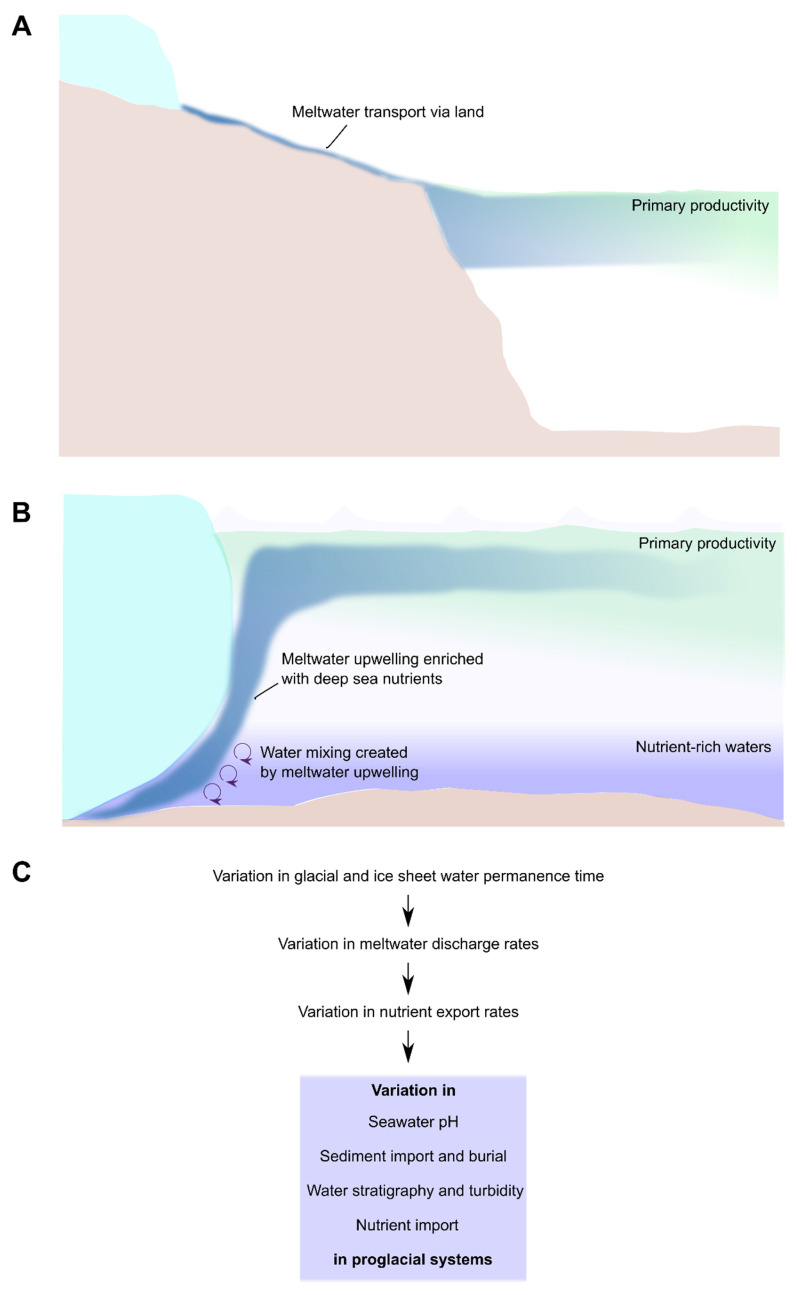
Land-terminating glaciers (**A**), marine-terminating glaciers (**B**), and main consequences of ice retreat of proglacial systems (**C**).

**Table 1 microorganisms-11-01153-t001:** Water availability and residence times in glacial/ice sheet compartments.

Location	Water Presence in Accumulation Season	Residence Time and Water Flow	Important References
Ice-lidded cryoconite holes	Yes, but some can completely freeze during accumulation season	Years, but occasionally connected to the rest of the system during accumulation season. Estimates say that ~50% of the cryoconite holes hydrologically connect to the supraglacial system every accumulation season.	[61,91]
Open cryoconite holes	No	During accumulation season, they have a higher connectivity to the glacial system than ice-lidded cryoconite holes, showing lower permanence times.	[39,63]
Weathering crust	Yes, if insulated by a snowpack.	Several days, water is released to the glacial when the system is saturated. Water flow is slow in the interstitial space.	[55,58,60]
Supraglacial streams	No	Depending on the ablation stage of the system, water can flow at different rates.	[64]
Supraglacial ponds and lakes	Yes. Lakes can form ice lids and maintain water during accumulation season.	Years; water is released to the glacial when the system is saturated. Smaller ponds can be drained by meltwater streams.	[39,50,61,64]
Firn aquifer	Yes	Perennial; water is released to the glacial when the system is saturated.	[52,92,93,94,95]
Englacial conduits	Yes, when the presence of solutes and particles lowers water freezing temperature.	Fast-flowing systems with permanence of hours up to a day. However, water can be present all year round and even for multiple years if water pockets are formed by collapsed conduits. Depending on the ablation stage of the system, water can flow at different rates.	[88,96,97]
Ice veins	Yes	Residence times in the ice veins are unknown. However, due to the low water flowing rate, we assume it to be in the order of years (at the very least).	[98]
Subglacial cavities	Yes	Days to months, and water is usually present all year round.	[99,100,101]
Subglacial channels	No	Hours, water is usually present only during peak ablation season.	[99,100,101]
Subglacial till	Yes	Potentially all-year round.	[36,78]
Subglacial lakes	Yes	Years; some systems have estimated water residence times of millions of years.	[79,80,102]

**Table 2 microorganisms-11-01153-t002:** Cell concentration in different water environments. VLP: Viral-Like Particles.

Cell Concentration	Source	Sampling Ablation Season	Reference
1.0–4.5 × 10^4^ cells mL^−1^ (3.97–12.7 × 10^4^ VLP mL^−1^)	Cryoconite holes, Midtre Lovénbreen	2000 and 2001	[186]
1.38 × 10^4^–4.84 × 10^4^ cells mL^−1^	Supraglacial meltwater runoff and cryoconite holes, Midtre Lovénbreen	2004	[187]
5.4 ± 1.6 × 10^4^ cells mL^−1^	Cryoconite holes, Austre Broggerbreen	2005	[31]
3.4 ± 1.2 × 10^4^ cells mL^−1^	Cryoconite holes, Midtre Lovénbreen	2005	[31]
4.1 ± 3.8 × 10^4^ cells mL^−1^	Cryoconite holes, Rotmoosferner	2004	[31]
3.7 ± 1.4 × 10^4^ cells mL^−1^	Cryoconite holes, Stubacher Sonnblickkees	2007	[31]
1.3 ± 8.2 × 10^4^ cells mL^−1^	Cryoconite holes, blue ice close to Patriot Hills	2002	[31]
4.4 ± 2.4 × 10^4^ cells mL^−1^	Cryoconite holes, Canada, Commonwealth, and Taylor glaciers	2005	[31]
2 × 10^4^ cells mL^−1^	Supraglacial meltwater runoff, Midtre Lovénbreen	2010	[188]
8.38 × 10^3^ ± 9.85 × 10^3^ cells mL^−1^	Supraglacial meltwater runoff, Russell glacier	2012	[6]
2.2 × 10^4^ ± 5.5 × 10^4^ cells mL^−1^	Weathering crust, Northern Hemisphere glaciers	2014, 2015 and 2016	[29]
6 × 10^4^ cells ml^−1^	Subglacial brine, Blood Falls	2004	[174]
10^6^–10^8^ cells mL^−1^	Ice vein water	/	[154]
4.7–5.7 × 10^5^ cells mL^−1^	Subglacial water, Skaftá subglacial lake	2006	[189]
4.4 ± 2.2 × 10^5^ cells mL^−1^	Subglacial water, East Skaftárkatlar subglacial lakes	2007	[162]
1.3 × 10^5^ cells mL^−1^	Subglacial water, subglacial lake Whillans	2013	[190]
1.3 × 10^5^ cells mL^−1^	Subglacial water, subglacial lake Whillans	2013	[191]
1.15 × 10^5^ ± 1.38 × 10^5^ cells mL^−1^	Subglacial meltwater runoff, Leverett glacier	2012	[6]

## Data Availability

No new data were created or analyzed in this study. Data sharing is not applicable to this article.

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
