# Peer review of "Glacial Water: A Dynamic Microbial Medium"

_microorganisms, 2023, doi:10.3390/microorganisms11051153_

Round 1
Reviewer 1 Report
The paper is well-described and sufficiently discussed. I have only a few minor questions and remarks.
1.
Lines 48-49: "Although several recent reviews of the microbiomics of glacial systems are available [4,30,31], here we focus on..."
You have quoted Anesio et al. npj Biofilms and Microbiomes (2017), Boetius et al. Nature Reviews Microbiology (2015), and Hotaling et al. Environmental Microbiology (2017).
Please, describe briefly the contribution of this review to the current state of knowledge, i.e. already described in the abovementioned reviews. In the other words, please describe the novelty of this review against other papers.
2.
Line 229: "residence times increase dramatically".
What is "dramatically"? Please, specify it or give an example.
3.
Table 2.
I think the data in table 2 should be presented as a box plot graph. Then the differences in values and standard deviations will be more readable.
4.
Lines 553-555: "All these data will provide more precise input data for models outlining proglacial (marine or land) shifts caused by climate change and the consequent change in meltwater dynamics".
Please, describe the predicted impact of climate change / global warming on the microbial communities in glacial water and its dynamics.
If "all these data will provide more precise input data for models", I deduce, you consider existing models as "non-sufficient". Please, discuss this issue.
Reviewer 2 Report
Dear authors,
Here is my review of the Manuscript ID: microorganisms-2315088, entitled: Glacial water: a dynamic microbial media, by Gilda Varliero et al. This review manuscript on the integration of microbial activity and nutrient and carbon dynamics concerning glacial water is a combination of interesting data and in my opinion it can be published in the Microorganisms journal after a minor revision.
Specific comments
Although the aim of the manuscript is to highlight the interdependence between microbial activity and glacial hydrology, any information to explain the microbial activity like at lines 30-32; 4. Glaciers and ice sheets as bioreactors, is missing from the Introduction.
Please, explain in a few sentences how microbial communities adapted to these icy environments can transform meltwater chemistry including both organic and inorganic compounds, making glaciers and ice sheets low-temperature bioreactors, before considering the dynamics of glacial ice sheet systems.
Any information with respect to the methodology of that type of investigation is missing too.
Figure 1. Reference(s) is need here.
Best wishes
